# A Distributed and Privacy-Preserving Random Forest Evaluation Scheme with Fine Grained Access Control

Yang Zhou [1,*], Hua Shen [2] and Mingwu Zhang [2]

1 School of Computer Science and Artificial Intelligence, Wuhan University of Technology, Wuhan 430070, China

2 School of Computers, Hubei University of Technology, Wuhan 430068, China; cshshen@hbut.edu.cn (H.S.); csmwzhang@gmail.com (M.Z.)

* Correspondence: zhouyang_cs@whut.edu.cn

**Abstract:** Random forest is a simple and effective model for ensemble learning with wide potential applications. Implementation of random forest evaluations while preserving privacy for the source data is demanding but also challenging. In this paper, we propose a practical and fault-tolerant privacy-preserving random forest evaluation scheme based on asymmetric encryption. The user can use asymmetric encryption to encrypt the data outsourced to the cloud platform and specify who can access the final evaluation results. After receiving the encrypted inputs from the user, the cloud platform evaluates via a random forest model and outputs the aggregated results where only the designated recipient can decrypt them. Threat analyses prove that the proposed scheme achieves the desirable security properties, such as correctness, confidentiality and robustness. Moreover, efficiency analyses demonstrate that the scheme is practical for real-world applications.

**Keywords:** privacy-preserving; robustness; fine grained access control; random forest; ensemble learning

## 1. Introduction

Nowadays, data evaluation using Machine Learning (ML) has been used in many real-world applications, such as spam classification[1], credit risk evaluation [2], medical-aided [3,4], etc. People are using smart services more [5], and more AI-related papers are being published [6]. This is because the evaluation results obtained by the users or data analysts can be used to provide assistance for management and decision-making.

Generally speaking, it is desirable to derive a stable and well-performing model directly after training a large amount of data. However, it is non-trivial in practice, because these data are considered as a digital asset. In many circumstances, it is difficult to collect enough data to train a well-performing machine learning model, and these collected data may show homogeneity, which may also make the trained model not generalize well. Therefore, the model generated by ML may not be satisfactory for the evaluation purpose.

Ensemble Learning [7] can be used to alleviate the above problem by integrating multiple weak models into one with better quality. Although a weak model may generate an unsatisfactory prediction, the other models can be used to balance the distortion (Note that one needs to optimize the training phase in order to ensure that weak models can indeed be combined into stronger ones, e.g., using the idea of Bagging). Suppose a patient is suffering some serious diseases, it is normal practice for several doctors with different experiences to diagnose together so that they can get a better overall view of the patient's health condition. The above method that uses group intelligence is very similar to the idea of Ensemble Learning. Random forest [8] is a typical algorithm in Ensemble Learning that contains many decision trees. Each of these decision trees evaluates the data individually during prediction, and random forest decides which category the data belongs to by running voting among these trees. Random forest also has similar limitations that require further optimization when combining the decision trees. In this work, we assume that

such optimizations have been carried out in the training phase. The random forest enjoys a number of attractive properties, such as high performance, good adaptation and allowance for parallel processing.

Recently, people are paying great attentions on user privacy, and many privacy regulations (e.g., Health Insurance Portability and Accountability Act (HIPAA) [9] in the US and General Data Protection Regulation (GDPR) [10] in the European Union) have been issued worldwide, requiring the service providers in ML to protect user privacy when offering services. If evaluation services cannot be provided on a privacy-protective basis, then they may not only be illegal, but also lose their appeal to privacy-conscious people, especially in the medical field [11]. Moreover, the trained models in ML are also valuable intellectual assets for the service providers, and they are unwilling to disclose these models as this may decrease their competitive advantages. Asymmetric encryption is the main building block to achieve privacy-preserving computations [12,13]. Compared with symmetric encryption, it can achieve homomorphic operations on ciphertext, which can solve the problem of privacy-preserving data sharing, eliminate data silos and promote the effective use of data.

### 1.1. Our Contributions

To solve the above issue, we propose a distributed privacy-preserving random forest evaluation scheme that achieves fine-grained access control by using asymmetric encryption. The protocol employs some novel cryptographic primitives and it provides accurate evaluation results without leaking either users' input data or service providers' trained models. Moreover, the evaluation results are only available to the designated recipients but no one else.

In the proposed scheme, the user can be offline after submitting the encrypted data. Once the evaluation is complete, the cloud platform can re-encrypt the results so that only the designated recipients can decrypt it. Moreover, the scheme is robust, i.e., even if a few servers drop out of the protocol due to temporary network problems, the rest of the servers can continue to execute the protocol.

Our work has potential applications in many scenarios and we use the following example to highlight our motivation. Suppose a user applies for loans from a bank, and the bank wants to conduct a comprehensive credit risk evaluation for this user. The bank can incorporate with multiple financial institutions, such as some other banks and insurance companies, to evaluate the user's data so that it can obtain comprehensive information to decide whether to provide loans for this user.

### 1.2. Related Works

In privacy-preserving random forest evaluation, existing works mainly focus on the privacy of base classifier within the decision tree algorithm. By constructing basis blocks, the work in [14] implements a privacy-preserving scheme for three algorithms, including decision trees. This work has demonstrated that polynomials can be used to represent decision trees. One needs to compare the node values of the decision tree with the evaluation data, and then uses the outputs to calculate the polynomial, obtaining the evaluation results. For example, the decision tree in Figure 1 predicts that the result polynomial can be represented as $\mathbf{P} = (1 - b_1)((1 - b_2)z_1 + b_2 z_1) + b_1(b_3 z_4 + (1 - b_3)((1 - b_4)z_3 + b_4 z_2))$. The main constructing blocks used in designing the protocol is fully homomorphic encryption (FHE) [15,16]. However, this scheme is impractical as it requires a lot of interactions between the participants.

The work in [17] proposes a method to compare the results of decision node values with the help of the client. The privacy of the decision tree is achieved using Paillier encryption [18] and oblivious transfer (OT). When the scheme is extended to random forest evaluation, the server uses random values to mask the results in order to hide each model of decision tree as well as the evaluation results. The final aggregation eliminates all these random values to obtain the random forest evaluation results.

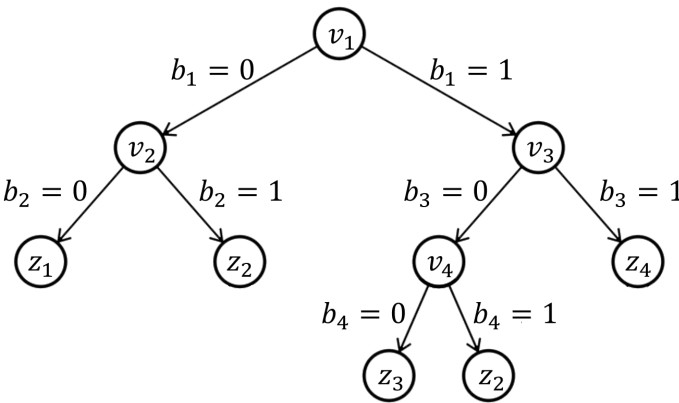

**Figure 1.** Decision tree model.

The work in [19] proposes a method to convert a decision tree into a series of linear equations represented by paths; the complexity of the evaluation is linearly related to the amount of nodes in the decision tree. Therefore, it is suitable for deep and sparse decision tree models. For example, the linear equation of decision tree paths in Figure 1 can be expressed as $z_1 = b_1 + b_2, z_2 = b_1 + (1 - b_2)$ or $z_2 = (1 - b_1) + b_3 + (1 - b_4), z_3 = (1 - b_1) + b_3 + b_4, z_4 = (1 - b_1)(1 - b_3)$. This protocol is designed using additive homomorphic encryption, that reduces the number of interactions compared with the other relevant schemes. Using the path equation and the OT protocol, the user can obtain the final classification results. It has been suggested that the scheme can be extended to support privacy-preserving evaluation of random forest, while the number of interactions can remain unchanged. That is, the decision trees can be executed in parallel and the processed information can be sent together for each interaction.

The work in [20] uses a commodity-based model to construct a two-party protocol in which the authority sends relevant data to the participants in the initial stage. The authority does not need to be involved in the subsequent execution of the protocol. The functionality of the authority can be pre-computed, and this improves efficiency. This protocol uses secret sharing as the main construct block to implement privacy-preserving classification of decision trees, etc. However, it lacks the ability to protect full information of the decision tree model and it does not support users to be offline.

For most interactive privacy-preserving decision tree protocols, only the client and the server are required to participate. Therefore, the structure is simple and it is easy to implement. However, it requires the client to remain online. However, without adding virtual decision tree nodes, it may leak information about the number of nodes or depth of the decision tree through the number of interactions between these two parties.

Nowadays, it is desirable to extend decision trees to random forest, and some researchers have investigated privacy-preserving random forest evaluation. In [21], each model owner sends her model in the encrypted form to an evaluator, and the user sends the encrypted data to this evaluator. Using the Multi-Key BGV scheme [22,23], the evaluator computes on the ciphertexts and performs random forest evaluation. The final result needs to be processed by all model owners before it is decrypted. Note that if the evaluator or any model owner fails during this period, the user may not be able to decrypt the result. We have to note that the majority of the existing FHE-based schemes are not suitable in practice, because FHE occurs heavy computational overheads and high storage costs [24]. Most existing non-interactive privacy protection schemes are based on asymmetric encryption (public key encryption with homomorphic properties), and we will also use this technology.

### 1.3. Organization

The notations and technical concepts are presented in Section 2. Section 3 describes the system model, security model and design goals. In Section 4, we first outline the constructing blocks for the proposed scheme and then introduce the proposed privacy-preserving random forest evaluation scheme. Security and efficiency analyses are given in Sections 5 and 6, respectively. Finally, in Section 7, we summarized the work of this paper and discussed the future work.

## 2. Preliminaries

In this section, we will introduce some preliminaries that will be used in the proposed scheme. The notations and abbreviations used in the paper are listed in Table 1.

**Table 1.** Abbreviations and notations.

| Symbols | Description |
| --- | --- |
| SR | Service requester |
| RR | Result recipient |
| CP | Cloud platform |
| ESPs | Evaluation service providers |
| $t$ | Number of servers in the cloud platform |
| $pk_{rr}/sk_{rr}$ | The key pair of RR |
| $PK/SK$ | The key pair of cloud platform |
| $[m]$ or $(A, B)$ | Encryption of $m$ under $PK$ |
| $\langle SK \rangle_i$ | $ESP_i$'s secret share of $SK$ |
| $\|m\|$ | Bit length of $m$ |
| $|\mathcal{I}|$ | Number of elements within $\mathcal{I}$ |
| $m_i$ | The raw data provided by SR |
| **P** | Polynomial expression of decision tree |
| $v_i$ | The value of the non-terminal node in the tree |
| **DCP** | Distributed comparison protocol |
| **DMUP** | Distributed multiplication protocol |
| **DMAX** | Distributed maximum protocol |
| **DMAX_n** | Distributed maximum_n protocol |
| **DRE** | Distributed re-encryption protocol |

### 2.1. Decision Tree

Decision trees are a non-parametric supervised learning algorithm with a wide range of applications that deals with nonlinear features, and its decision rules can be easily explained. In this paper, we use the binary decision trees, for example, classification and regression trees (CART), denoted as $\mathcal{T}(\mathcal{V}, \mathcal{Z})$. The non-leaf nodes are called decision nodes and we assume that there are $\sigma$ of these nodes, each containing a value $v_i \in \mathcal{V}(i \in \{1, 2, \ldots, \sigma\})$. Each branch represents the testing result of the decision node with respect to the value of the data $m_j$ to be evaluated. Let $e_i = 1$ represents $m_j < v_i$ and the left branch is chosen, while $e_i = 0$ denotes $m_j \geq v_i$ and the right branch is chosen. The $(\sigma + 1)$ terminal nodes represent the $|\mathcal{Z}| = \delta$ categories. When a decision tree model is used for evaluation, the comparison starts from the root node, and the branch is selected based on the comparison result, and then the comparison continues for the nodes on the branch until the leaf node is obtained, and the category $z_k \in \mathcal{Z}(k \in \{1, 2, \ldots, \delta\})$ means that the node is taken as the decision evaluation.

### 2.2. Ensemble Learning and Random Forest

Although decision trees have many advantages, the models obtained from training are susceptible to overfitting by the data set. Therefore, ensemble learning, which ensembles multiple models in an appropriate way to improve the evaluation performance has emerged. Ensemble learning algorithms mainly include the parallel Bagging algorithm and the serial Boosting algorithm.

The Random forest algorithm proposed by Breiman [8] is one of the most representative and top performing algorithms among the bagging methods. It has been applied to different tasks [2,25] due to its simple parameters and high adaptability. Random forest builds multiple decision trees in the training phase and ensembles them to obtain more accurate and stable prediction results. In the evaluation phase, the data are inputted into each decision tree for evaluation individually, and the final evaluation results are obtained by voting among the decision trees. We have to note that the privacy-preserving evaluation scheme studied in this paper is based on plurality voting, which means that the output of each model is treated as one vote, and the prediction is taking the one with most votes, and ties are broken arbitrarily. Random forest address the performance bottleneck of decision tree, and they have better tolerance to noise. Moreover, they can be executed in parallel.

*2.3. Secret Sharing Scheme*

Secret sharing divides a secret $s$ into $t$ secret shares by the secret sharing algorithm, and sends the shares to $t$ participants. When we want to recover $s$, we need a quorum of the participants to use their shares to reconstruct the secret $s$. If $(k, t)$ threshold is used, it needs $k$ participants to complete the secret reconstruction algorithm, while less than $k$ participants know nothing about the the secret $s$. It is not necessary for everyone to participate in each reconstruct, which makes the scheme participants who use the threshold more flexible. In this paper, we use the Shamir secret sharing scheme [26].

The Shamir secret sharing scheme is a $(k, t)$ threshold scheme, which has a secret $s$ to be shared and $t$ participants. There are mainly two algorithms:

**Sharing**$(s, k, x_1, x_2, \ldots, x_t)$: The secret sharing algorithm first selects $k - 1$ random numbers $(a_1, a_2, \ldots, a_{k-1})$ from $\mathbb{Z}_p$, and uses these random numbers to construct a polynomial of degree $k - 1$: $f(x) = s + \sum_{i=1}^{k-1} a_i x^i \pmod{p}$, where $x_j (j \in \{1, 2, \ldots, t\})$ and the polynomial is evaluated as $f(x_j)$. $x_j$ are public values associated with the participants. To simplify the description, we set $x_i = i$.

**Recon**$(\{\langle s \rangle_i, x_i\}_{(i \in \mathcal{I})}, k)$: The secret reconstruction algorithm first verifies $|\mathcal{I}| \geq k$, if not, it terminates. Otherwise, it uses the Lagrange interpolation formula to calculate $s = \sum_{i \in \mathcal{I}} (\langle s \rangle_i \prod_{j \in \mathcal{I}, j \neq i} \frac{x_j}{x_j - x_i})$

In addition, the scheme satisfies the homomorphic property that if $c = a + b$ and $x_1, x_2, \ldots, x_t$ is consistent, then $\langle c \rangle_i = \langle a \rangle_i + \langle b \rangle_i$.

*2.4. Distributed BCP Cryptosystem with Threshold Decryption*

The BCP cryptosystem [27] is an asymmetric cryptosystem, which we use as the primary encryption scheme to construct our proposed privacy-preserving random forest evaluation system in a distributed fashion. In order to match the system structure, we modified the original scheme in order to support $(k, t)$ threshold decryption. This modified algorithm is called Distributed Threshold Re-Encryption Scheme (DTRS) and mainly includes the following algorithms.

**Setup**$(\kappa)$: The Setup algorithm generates the public parameters $pp$ according to the security parameter $\kappa$. Two large strong primes $p, q$ are randomly selected, satisfying $|p| = |q| = \kappa$. Strong prime numbers require that $p, q$ have the form of $p = 2p' + 1$ and $q = 2q' + 1$, where $p'$ and $q'$ are also primes. Then, $N = pq$. $\mathbb{G}$ is the cyclic group of quadratic residues modulo $N^2$ and randomly selecting $g$ to satisfy the maximum order is $ord(\mathbb{G}) = pp'qq'$. The public parameters $pp$ are $N, g$.

**KeyGen**$(pp)$: Key generation algorithm generates the public and private key pairs for the users according to $pp$. The users randomly select a number $u_i \in \mathbb{Z}_{N^2}^*$ as the private key $sk_i$ and calculates $g^{u_i}$ as the corresponding public key $pk_i$.

**Enc**$(m, PK)$: Given a plaintext $m \in \mathbb{Z}_N$, the ciphertext is generated using the encryption algorithm and the public key $PK$. One first chooses a random number $r \in \mathbb{Z}_N^*$, and then the ciphertext $(A, B)$ is computed as $A = (1 + m \cdot N)PK^r, B = g^r$.

**Dec**$((A, B), SK)$: With the knowledge of $SK$, $m$ can be obtained as follows: $m = L(A/(B)^{SK} \pmod{N^2}) \pmod{N}$, where $L(x) = \frac{x-1}{N}$.

**PDec**$(B, \langle SK \rangle_i)$: When the private key is secretly shared, the partial decryption algorithm is executed according to the share of the private key. $B^{(i)}$ can be calculated as: $B^{(i)} = B^{2\Delta \langle SK \rangle_i} \pmod{N^2}$.

**CDec**$(A, \{B^{(i)}\}_{i \in \mathcal{I}})$ When $|\mathcal{I}|$ is greater than $k$, i.e., no less than $k$ copies of $B^{(i)}$ are received, the plaintext $m$ can be obtained by performing the combining algorithm. $m = L(A^{2\Delta} / \prod_{i \in S}(B^{(i)})^{\mathcal{L}_i(0)} \pmod{N^2})/2\Delta \pmod{N}, \Delta = t!$.

Remark: Here, $\Delta = t!$ is introduced because it is infeasible to perform the inverse operation when computing Lagrange interpolation on the exponent while executing **CDec**. The solution is given in [28], which avoids computing the inverse element by multiplying by $\Delta$.

**REnc**$((A, B), pk_{rr}, SK)$: If we have the private key $SK$ of ciphertext $(A, B)$, one can re-encrypt the ciphertext $(A, B)$ into another ciphertext $(\widetilde{A}, \widetilde{B})$ without decryption. Only the person who has the private key corresponding to the public key $pk_{rr}$ can decrypt the ciphertext after re-encryption. The ciphertext after re-encryption is calculated as $\widetilde{A} = A, \widetilde{B} = (B \cdot pk_{rr})^{SK}$. Obviously, re-encryption needs to be performed by multiple parties, otherwise it has the same effect as decryption first and encryption later, which makes no sense.

**RDec**$\left(\left(\widetilde{A}, \widetilde{B}\right), sk_{rr}\right)$: The private key $sk_{rr}$ can decrypt the re-encrypted ciphertext $\left(\widetilde{A}, \widetilde{B}\right)$, $m$ can be obtained as follows: $m = L((\widetilde{A} \cdot PK^{2\Delta \cdot sk_{rr}})/\widetilde{B} \pmod{N^2})/2\Delta \pmod{N}$, where $L(x) = \frac{x-1}{N}$.

**CM**$([m])$: Ciphertext can be modified without changing the corresponding plaintext. Choose a number $r' \in \mathbb{Z}_N^*$ at random, and the new ciphertext can calculate by $[m]_{new} = (1 + m \cdot N)pk^r \cdot pk^{r'}, g^r \cdot g^{r'}$

Furthermore, the scheme has the following properties:

- 
$$
\begin{aligned}
[m]^k &= (((1 + m \cdot N)pk^r)^k, (g^r)^k) \bmod N^2 \\
&= ((1 + k \cdot m \cdot N)pk^{rk}, g^{rk}) \bmod N^2 \\
&= [k \cdot m]
\end{aligned}
$$

- 
$$
\begin{aligned}
[m]^{N-1} &= (((1 + m \cdot N)pk^r)^{(N-1)}, (g^r)^{(N-1)}) \bmod N^2 \\
&= ((1 + (N-1)mN)pk^{rN-r}, g^{rN-r}) \bmod N^2 \\
&= [-m]
\end{aligned}
$$

- Additive homomorphism: If $m = \sum_{i=1}^n m_i$, then $[m]$ can be calculated by $[m_i]$

$$
[m] = (A, B) = \prod_{i=1}^n [m_i] = \prod_{i=1}^n (A_i, B_i)
$$

## 3. Models and Definitions

### 3.1. System Model

Our proposed privacy-preserving random forest evaluation scheme is constructed using the system model as shown in Figure 2. There are three different types of entities in the system we designed: Evaluation Service Providers (ESPs), Service Requestor (SR) and Results Recipient (RR).

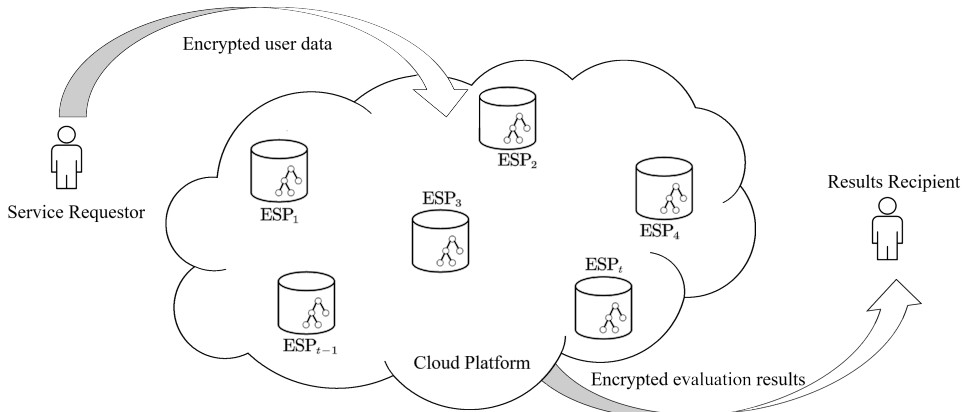

**Figure 2.** System model.

### 3.1.1. Evaluation Service Providers (ESPs)

For external entities, all ESPs collaborate to form a cloud platform (CP), providing random forest evaluation services. These ESPs are structured in a distributed fashion, and they evaluate the encrypted data on their own models. The final results are re-encrypted and sent to the RR.

Suppose there are $t$ ESPs in CP, each ESP possesses some decision tree models that are base learners of the random forest. $\mathcal{T}_{ij}(\mathcal{V}, \mathcal{Z})$ denotes the $j$-th($j \in \{1, 2, \ldots, o_i\}$) decision tree model for the $i$-th($i \in \{1, 2, \ldots, t\}$) ESP. To simplify the description, it is assumed that $ESP_i$ has only one counterpart model $\mathcal{T}_i$. In reality, ESP may have more models, such complicated cases can be composed of the simple cases with one model.

### 3.1.2. Service Requestor (SR)

SR encrypts the information needed for the evaluation and transmits the ciphertext to the CP. Their purpose is to get the cloud platform's random forest evaluation service for their data. In this process, SR does not want their data or the evaluation results to be leaked. For user data, they are discretized and hashed into binary values, we suppose that this operation is public and the result is the same for the same data.

### 3.1.3. Results Recipient (RR)

RR is the designated entity to receive the evaluation results. It is determined according to the specific use scenario, this entity can be the SR himself, or someone who can provide guidance to SR based on the evaluation results, such as a doctor. Or it could be an institution that is ready to accept SR's business, such as bank, private doctor, etc.

### *3.2. Threat Model*

In the proposed scheme, all entities need to be honest-but-curious. They will strictly follow the protocols, but may record various data during the execution and intend to derive information about the other entities. An external adversary $\mathcal{A}$ is considered, that can listen to all exchanged messages. The data of the SR, the model of ESPs and the final evaluation results are all elements that the adversary $\mathcal{A}$ wants to obtain.

ESPs are assumed in a competitive relationship with each other. Rationally, they would incorporate to carry out the task but refuse to collude, because it may leak their machine learning models, affecting their competitive advantages.

### *3.3. Security Requirements*
#### 3.3.1. Correctness

If all participants follow the protocol, the cloud platform can accurately perform the evaluation of the SRs' data.

### 3.3.2. Confidentiality

SR and RR cannot obtain any information about the machine learning model of ESPs. ESPs learn nothing about data of SR and evaluation results. Each ESP's machine learning model should not be exposed to the other ESPs.

### 3.3.3. Flexibility

The evaluation results of the cloud platform can be sent to any designated RR; In practice, SRs and RR normally have only restricted computational power. They can remain offline after the SR uploads the encrypted data and before the RR receives the re-encrypted result.

### 3.3.4. Robustness

If a few servers are unable to participate in the designed security protocol due to a temporary network failure, in order to maintain the efficiency of the implementation, the remaining servers can continue to execute the protocol.

## 4. The Proposed Scheme

### 4.1. Constructing Blocks

DTRS enjoys the additive homomorphic property, but it cannot implement functions such as multiplication and comparison between two plaintexts. To construct our proposed scheme under honest-but-curious conditions, we modify the protocols to support comparison, multiplication, and re-encryption as in [29–31].

### 4.1.1. Distributed Multiplication Protocol

The protocol Algorithm 1 is designed to calculate the product of two original data, but in order to ensure that the protocol can be executed correctly in the final scheme, i.e., it is necessary to consider that the result remains in the plaintext space of the encryption scheme after multiple multiplications ($m = \prod_{i=1}^{a} m_i$). Therefore, it is necessary to limit the length of each raw data to satisfy the $\|N\|/2a > \|m_i\|$.

---

**Algorithm 1** Distributed multiplication protocol (**DMUP**).

---

**Input**: $ESP_\alpha$ gives two ciphertexts $[a], [b]$ encrypted with $PK$; $ESP_i$ own the key sharing share $\langle SK \rangle_i$; Public sharing parameters $x_1, x_2, \ldots, x_t$.
**Output**: $ESP_\alpha$ obtains $[a \cdot b]$.

1: $ESP_\alpha$ randomly chooses two numbers $r_a \in \mathbb{Z}_N$ and $r_b \in \mathbb{Z}_N$, computes $(A_a, B_a) = [a] \cdot [r_a] = [a + r_a]$; $(A_b, B_b) = [b] \cdot [r_b] = [b + r_b]$.
2: $ESP_\alpha$ sends $A_a$ and $A_b$ to $ESP_\beta (\beta \in \{1, 2, \ldots, t\} \backslash \alpha)$, $B_a$ and $B_b$ to all $ESP_i (i \in \{1, 2, \ldots, t\} \backslash \{\alpha, \beta\})$.
3: When $ESP_i$ receives the $B_a$ and $B_b$, they use $SK_i$ to calculate the $B_a^{(i)} = \textbf{PDec}(B_a, \langle SK \rangle_i), B_b^{(i)} = \textbf{PDec}(B_b, \langle SK \rangle_i)$. Then send the results to $ESP_\beta$.
4: When $ESP_\beta$ receives no less than $k$ copies of the $B_a^{(i)}$ and $B_b^{(i)}$, calculate $(a + r_a) = \textbf{CDec}(A_a, \{B_a^{(i)}\}_{i \in S_1}), (b + r_b) = \textbf{CDec}(A_b, \{B_b^{(i)}\}_{i \in S_2})$. Then, $ESP_\beta$ calculates $Z = (a + r_a)(b + r_b)$, encrypts $Z$ using $PK$, and sends $[Z]$ to $ESP_\alpha$.
5: Once $[Z]$ is received, $ESP_\alpha$ computes $[Z_1] = [a]^{(N-r_b)}$, $[Z_2] = [b]^{(N-r_a)}$ and $[Z_3] = [r_a \cdot r_b]^{(N-1)}$. It can be easily verified that $[a \cdot b] = [Z] \cdot [Z_1] \cdot [Z_2] \cdot [Z_3] = [(a + r_a)(b + r_b) - a \cdot r_b - b \cdot r_a - r_a \cdot r_b]$

---

As in the proposed PPRE scheme the product of multiple zeros or ones is calculated, the result is always within the required range.

### 4.1.2. Distributed Comparison Protocol

The purpose of this operation Algorithm 2 is to compare the size of two raw data $a$ and $b$. To ensure proper execution of the protocol, it is necessary to limit the length of the original data to be less than $\frac{\|N\|}{4} - 1$.

---

**Algorithm 2** Distributed comparison protocol (**DCP**).

---

**Input**: $\text{ESP}_\alpha$ gives two ciphertexts $[m_1], [m_2]$ encrypted with $PK$; $\text{ESP}_i$ own the key sharing share $\langle SK \rangle_i$; Public sharing parameters $x_1, x_2, \ldots, x_t$.

**Output**: $\text{ESP}_\alpha$ obtains $[e]$, We remark that $e = 0$ indicates $m_1 \geq m_2$, and $e = 1$ indicates $m_1 < m_2$.

1: $\text{ESP}_\alpha$ randomly chooses $\sigma \xleftarrow{R} \{0, 1\}$ and an integer $r \in \mathbb{Z}_N$, s.t. $\|r\| < \frac{\|N\|}{4}$ and $r \neq 0$, If $\sigma = 1$, then $\text{ESP}_\alpha$ computes

$$[m]_{1-2} = [m_1 - m_2] = (A_1 \cdot A_2^{N-1}, B_1 \cdot B_2^{N-1})$$

$$[l] = [r(2m_{1-2} + 1)] = ([m_{1-2}]^2 \cdot [1])^r = (A_l, B_l)$$

Otherwise, $\text{ESP}_\alpha$ computes

$$[m]_{2-1} = [m_2 - m_1] = (A_2 \cdot A_1^{N-1}, B_2 \cdot B_1^{N-1})$$

$$[l] = [r(2m_{2-1} + 1)] = ([m_{2-1}]^2 \cdot [1])^r = (A_l, B_l)$$

2: $\text{ESP}_\alpha$ sends $A_l$ and $B_l$ to $\text{ESP}_\beta (\beta \in \{1, 2, \ldots, t\} \setminus \{\alpha\})$ and all $\text{ESP}_i (i \in \{1, 2, \ldots, t\} \setminus \{\alpha, \beta\})$ respectively.

3: When $\text{ESP}_i$ receives the $B_l$, they use $B_l$ and $\langle SK \rangle_i$ to calculate the $B_l^{(i)} = \mathbf{PDec}(B_l, \langle SK \rangle_i)$. Then send the results to $\text{ESP}_\beta$.

4: When $\text{ESP}_\beta$ receives no less than $k$ copies of the $B_l^{(i)}$, calculate $l = \mathbf{CDec}(A_l, \{B_l^{(i)}\}_{i \in \mathcal{I}})$. If $\|l\| > \frac{\|N\|}{2}$, then $\text{ESP}_\beta$ orders $e^* = 1$, otherwise $e^* = 0$. $\text{ESP}_\beta$ encrypts $e^*$ using the **Enc** and transmits the ciphertext to the $\text{ESP}_\alpha$.

5: Once $[e^*]$ is received, if $\sigma = 1$, $\text{ESP}_\alpha$ invokes **CM** to compute $[e] = \mathbf{CM}([e^*])$, otherwise compute $[e] = [1] \cdot [e^*]^{N-1} = [1 - e^*]$.

---

### 4.1.3. Distributed Maximum Protocol

This protocol is Algorithm 3 similar as the above one where the larger number is returned after comparing two numbers.

---

**Algorithm 3** Distributed maximum protocol (**DMAX**).

---

**Input**: $\text{ESP}_\alpha$ gives two ciphertexts $([num_1], [z_1])$, $([num_2], [z_2])$ encrypted with $PK$; $\text{ESP}_i$ own the key sharing share $\langle SK \rangle_i$; Public sharing parameters $x_1, x_2, \ldots, x_t$.

**Output**: $\text{ESP}_\alpha$ obtains the $([num_{max}], [z_{max}])$ corresponding to the maximum $num$ value.

1: $\text{ESP}_\alpha$ run **DCP**$([num_1], [num_2]) \rightarrow [e]$.

2: $\text{ESP}_\alpha$ calculates **DMUP**$([e], [num_2]) \cdot$ **DMUP**$([1] \cdot [e]^{N-1}, [num_1]) \rightarrow [num_{max}]$ and **DMUP**$([e], [z_2]) \cdot$ **DMUP**$([1] \cdot [e]^{N-1}, [z_1]) \rightarrow [z_{max}]$

---

### 4.1.4. Distributed Maximum_n Protocol

Invoking the previous protocol, one can compare multiple raw data and obtain the largest one Algorithm 4.

---

**Algorithm 4** Distributed maximum_n protocol (**DMAX_n**).

---

**Input**: $ESP_\alpha$ gives some ciphertexts $([num_1], [z_1]), ([num_2], [z_2]), \ldots, ([num_n], [z_n])$ encrypted with $PK$; $ESP_i$ own the key sharing share $\langle SK \rangle_i$; Public sharing parameters $x_1, x_2, \ldots, x_t$.

**Output**: $ESP_\alpha$ obtains the $[z_{max}]$ corresponding to the maximum $num$ value.

1:  $ESP_\alpha$ run **DMAX**$(([num_1], [z_1]), ([num_2], [z_2])) \rightarrow ([NUM], [CLASS])$.
2:  **for** $i = 3$ to $n$ **do**
3:     **DMAX**$(([NUM], [CLASS]), ([num_i], [z_i])) \rightarrow ([NUM], [CLASS])$.
4:  **end for**
5:  **return** $[z_{max}] = [CLASS]$.

---

### 4.1.5. Distributed Re-Encryption Protocol

Here, we implement proxy re-encryption in the distributed fashion Algorithm 5. The ciphertext encrypted with the $PK$ can be converted to the ciphertext encrypted with the given public key with the cooperation of multiple servers that have a share of the master private key $SK$.

---

**Algorithm 5** Distributed re-encryption protocol (**DRE**).

---

**Input**: $ESP_\alpha$ gives a ciphertext $(A, B)$ encrypted with $PK$; Public key $g^u$ of the assigned user; $ESP_i$ own the key sharing share $\langle SK \rangle_i$; Public sharing parameters $x_1, x_2, \ldots, x_t$.

**Output**: $ESP_\beta$ obtains the re-encrypted ciphertext $(\widetilde{A}, \widetilde{B})$ that can only be decrypted using the user's private key $u$.

1:  $ESP_\alpha$ computes $\bar{B} = B \cdot g^u$ and $\widetilde{A} = A^{2\Delta}$.
2:  $ESP_\alpha$ sends $\widetilde{A}$ to $ESP_\beta (\beta \in \{1, 2, \ldots, t\} \backslash \{\alpha\})$ and $\bar{B}$ to all $ESP_i (i \in \{1, 2, \ldots, t\} \backslash \{\alpha, \beta\})$.
3:  When $ESP_i$ receives the $\bar{B}$, they use $\langle SK \rangle_i$ to calculate the $\bar{B}^{(i)} = \mathbf{PDec}(B, \langle SK \rangle_i)$. Then send the results to $ESP_\beta$.
4:  When $ESP_\beta$ receives $\widetilde{A}$ and no less than $k$ copies of the $\bar{B}^{(i)}$, reconstructing $\widetilde{B}$ using $\{\bar{B}^{(i)}, i\}_{i \in \mathcal{I}}$. $ESP_\beta$ obtains the re-encrypted ciphertext $(\widetilde{A}, \widetilde{B})$.

---

### 4.2. Initialization

Our proposed solution is structured in a distributed fashion without a trusted third party (TTP), so all ESPs need to work together to generate the system parameters in a distributed manner before providing the evaluation services.

### 4.2.1. Model Training

Our scheme addresses privacy-preserving in the evaluation phase, so it does not impose any constraints on the training phase of the random forest, and the ESP can train the model using the method in [32], etc. Regardless of the method used by the ESP to obtain the model, we only require that it is compliant with the random forest. Having a trained decision tree model for each ESP will serve as a precondition for us to design a privacy-preserving random forest evaluation scheme.

### 4.2.2. Public Parameters

The setup phase can be executed using the method as in [33]. Parties agree on the following information:Modulo $N = pq$, $p, q$ are strong prime, but no one knows the exact value of $p, q$. $\mathbb{G}$ is the cyclic group of quadratic residues modulo $N^2$ and randomly selecting $g$ to satisfy the maximum order is $ord(\mathbb{G}) = pp'qq'$. The system parameters are $N, g, \mathbb{G}$. Meanwhile, the cloud platform can set the threshold $k$.

### 4.2.3. Cloud Platform Private Key Share and Public Key

$ESP_i(i = 1, 2, \ldots, t)$ randomly chooses different integers $sk_i \in \mathbb{Z}_{N^2}^*$, and the system private key $SK$ is defined as $\sum_{i=1}^{t} sk_i \pmod{N^2}$. Using the previous selection of $g$, one can calculate $PK = g^{SK}$ as the system public key.

ESPs use Shamir secret sharing to calculate their own selected shares of $sk_i$ and share them to all ESPs. After receiving shares from the other ESPs, $ESP_i$ holds $(\langle sk_1 \rangle_i, \langle sk_2 \rangle_i, \ldots, \langle sk_t \rangle_i)$, where $\langle sk_1 \rangle_i$ denotes the secret sharing share of $sk_1$. As the secret sharing algorithm has the additive homomorphic property, $ESP_i$ can calculate $\langle SK \rangle_i = \sum_{i=1}^{t} \langle sk_1 \rangle_i$ to get the sharing share of the master private key. At this moment, the cloud platform announces $(N, g, PK, t)$ to the public.

### 4.3. Privacy-Preserving Random Forest Evaluation(PPRE)

By using the above constructed block and DTRS, we give the privacy-preserving random forest evaluation scheme. In our scheme, the idea of expressing the decision tree as a polynomial **P** is used [14]. Using this idea, we need to compare the nodes of the decision tree with the corresponding data to be evaluated, but only the coefficients of the classification in the polynomial **P** are computed. Afterwards they are handed over to a server for aggregation and sorting to give the evaluation results of the random forest.

In our proposed PPRE algorithm, SR can go offline after sending the data to be evaluated and the public key to CP. After CP receives the evaluation request, the servers in the cloud cooperate to process the encrypted data and re-encrypt the processing result with the public key after aggregation. Finally, the re-encrypted ciphertext is given to RR. After RR receives the ciphertext and decrypts it, the evaluation result of SR's data can be obtained. The details are shown in the Table 2.

**Table 2.** Privacy-Preserving Random Forest Evaluation Scheme Design.

---

**Setup phase:**
**ESPs(CP):**
1. Perform the operations in Section 4.2 Initialization
2. Obtain a polynomial expression $\mathbf{P}_i$ for each decision tree model $\mathcal{T}_i(\mathcal{V}_i, \mathcal{Z}_i)$.
**SR:**
1. Obtain public information from the CP.
2. Generate data to be evaluated.
3. Select RR and forward information from the CP to RR
**RR:** Generate a public-private key pair $(sk_{rr} = u, pk_{rr} = g^u)$ and give the public key $pk_{rr}$ to SR.

---

**Phase1 Outsourcing:**
SR encrypts the data $\{m_1, m_2, \ldots, m_n\}$ with the public key $PK$ of the CP. Send the ciphertext $\{[m_1], [m_2], \ldots, [m_n]\}$ and RR's public key $pk_{rr}$ to the CP.

---

**Phase2 Evaluating:**
1. ESPs call **DCP** to compare the received ciphertext $m_j(j \in n)$ with the value of the corresponding node $v_k \in \mathcal{V}_i(k \in \{1, 2, \ldots, \sigma_i\})$ in its own decision tree to get the result $b_k$.
2. ESPs use **DMP** to compute the category coefficients of its own decision tree polynomials $\mathbf{P}_i$ and merges the coefficients of the same categories to obtain $\mathbf{P}_i = \{[coz_1] \cdot z_1, [coz_2] \cdot z_2, \ldots, [coz_\delta] \cdot z_\delta\}$.
3. ESPs select an $ESP_\gamma$ as the aggregation server and send the computed decision tree polynomial to it.

---

**Phase3 Aggregating:**
1. After receiving the decision tree polynomials results from all ESPs, $ESP_\gamma$ aggregates them using the additive homomorphism of DRTS to obtain $\{[soc_1] \cdot z_1, [soc_2] \cdot z_2, \ldots, [soc_\delta] \cdot z_\delta\}$.
2. $ESP_\gamma$ encrypts $z_1, z_2, \ldots, z_\delta$ and calls **DMAX_n** for sorting to get $[z_{max}](max \in (1, 2, \ldots T, \delta))$ which corresponds to the maximum $soc_{max}$.
3. $ESP_\gamma$ calls **DRE** re-encrypts $[z_{max}]$ using RR's public key, and the resulting ciphertext is sent to RR.

---

**Phase4 Decrypting:**
RR performs **RDec** decryption of the received ciphertext to obtain the CP's evaluation of the SR's data.

---

The evaluation results with the most votes from all ESPs will be provided at the end. If in practice more suggestions need to be provided, Tournament Sort can be constructed using block **DMAX** instead of **DMAX_n** protocol. After one sort, the top aggregated results can be sent to SR in order. Using this method, not only more information can be provided, but also efficiency can be improved by computing in parallel. Although Tournament Sort requires more storage space, the loss is insignificant compared to the benefits gained.

## 5. Security Analyses

### 5.1. Semantic Security of DTRS

We add a threshold mechanism to the encryption scheme to prove that the ciphertext can be decrypted correctly before the security is verified by the following equation:

$$B^{2\Delta \cdot SK} = \prod_{i \in \mathcal{S}}(B^{(i)})^{\mathcal{L}_i(0)} = \prod_{i \in \mathcal{S}}(g^{r \cdot 2\Delta \langle SK \rangle_i})^{\mathcal{L}_i(0)}$$

$$m = L((A^{2\Delta}/B^{2\Delta \cdot SK}) \pmod{N^2})/2\Delta \pmod{N}$$

**Theorem 1.** *If the DDH assumption over $\mathbb{Z}_{N^2}^*$ holds, the DTRS scheme satisfies the semantic security property.*

**Proof.** DTRS is changed based on the addition of a threshold mechanism to the BCP cryptosystem, which does not affect the semantic security of BCP. Obviously, if an adversary can break the semantic security of DTRS, then it is possible to use this adversary to break the semantic security of the BCP scheme. However, as the security of BCP is based on DDH assumption over $\mathbb{Z}_{N^2}^*$, then our encryption scheme is also secure.

The security of the private key being partitioned relies on the secret sharing scheme, and again, all sharing schemes used in this paper are proven to be information-theoretically secure. In addition, in the process of using the share, it only appears in the exponential part of the ciphertext and is not leaked out directly, any polynomial time adversary is unable to calculate the discrete logarithm directly to obtain the share. The original plaintext message can only be correctly restored after receiving not less than a threshold number of partial decryptions, which is guaranteed by the nature of threshold secret sharing. □

**Theorem 2.** *If the DDH assumption over $\mathbb{Z}_{N^2}^*$ holds, then the proposed Re-Encryption scheme in the DTRS is semantically secure.*

**Proof.** There is an original BCP ciphertext $A = (1 + m \cdot N)PK^r, B = g^r, PK = g^{sk}$. We assume that the public key of the specified recipient is $g^u$. Then, the ciphertext after re-encryption is $A = (1 + m \cdot N)PK^r, B = (g^r \cdot g^u)^{sk}$.

Obviously, a DH key exchange protocol is performed during the re-encryption process. Suppose an adversary $\mathcal{A}$ can know $g^{u \cdot sk}$ if $g^{sk}, g^u$ is specified, then he can break the security of re-encryption. Further, we can use this adversary $\mathcal{A}$ to solve the DDH assumption. □

### 5.2. Security of Multiplication Protocol

We adopt the security model to construct the ideal function against honest-but-curious adversaries. To simplify the description, we assume three participants $ESP_i$, $ESP_\alpha$ and $ESP_\beta$ are involved. First, we construct three simulators $Sim = (Sim_{E_i}, Sim_{E_\alpha}, Sim_{E_\beta})$ to emulate the adversaries $(\mathcal{A}_{E_i}, \mathcal{A}_{E_\alpha}, \mathcal{A}_{E_\beta})$ that corrupt $ESP_i$, $ESP_\alpha$ and $ESP_\beta$, respectively. The security proof of the multiplication protocol is as follows:

$Sim_{E_\alpha}$ simulates $\mathcal{A}_{E_\alpha}$ as follows: Randomly choose two numbers $\check{x}$ and $\check{y}$ in $\mathbb{Z}_N$ and encrypt them with **Enc** to get $[\check{x}]$ and $[\check{y}]$. After that, $Sim_{E_\alpha}$ sends the ciphertext $[\check{x}]$ and $[\check{y}]$ to $\mathcal{A}_{E_\alpha}$. After receiving the ciphertext, if $\mathcal{A}_{E_\alpha}$ terminates, then $Sim_{E_\alpha}$ terminates as well. In both real and ideal environments, the perspective of $Sim_{E_\alpha}$ is indistinguishable due to the semantic security of DTRS.

$Sim_{E_i}$ simulates $\mathcal{A}_{E_i}$ as follows: $Sim_{E_i}$ randomly generates two numbers $x_r$ and $y_r$ in $\mathbb{Z}_N^*$, and calculates $B_x = g^{x_r}$, $B_y = g^{y_r}$. Then, use **PDec** to obtain $B_x{}^{(i)}$ and $B_y{}^{(i)}$. Next, send these partially decrypted ciphertext to $\mathcal{A}_{E_i}$. If $\mathcal{A}_{E_\alpha}$ terminates, then $Sim_{E_\alpha}$ terminates as well. The perspective of $\mathcal{A}_{E_i}$ is ciphertext obtained using DTRS encryption. In both the real and the ideal environments, he receives the output $B_x{}^{(i)}$ and $B_y{}^{(i)}$. Securing in a real-world environment with the PDL Problem. The perspective of $\mathcal{A}_{E_i}$ is indistinguishable in either the ideal or real environment.

$Sim_{E_\beta}$ simulates $\mathcal{A}_{E_\beta}$ as follows: It gets $[x']$ by encrypting a random number $x' \in \mathbb{Z}_N$ using **Enc**, and then sends $[x']$ to $\mathcal{A}_{E_\beta}$. If $\mathcal{A}_{E_\alpha}$ terminates, then $Sim_{E_\alpha}$ terminates as well. The perspective of $\mathcal{A}_{E_\beta}$ is ciphertext obtained using DTRS encryption. In both the real and the ideal environments, he receives the output encryption $[x']$. Securing in a real-world environment with the DTRS. The perspective of $\mathcal{A}_{E_\beta}$ is indistinguishable in either the ideal or real environment.

Note that the security proofs of the Comparison, Re-Encryption protocols and so on are similar as in the Multiplication protocol against honest-but-curious adversaries $\mathcal{A} = (\mathcal{A}_{E_i}, \mathcal{A}_{E_\alpha}, \mathcal{A}_{E_\beta})$.

*5.3. Security of Random Forest Evaluation*

First, SR encrypts the data and sends them to the CP. As the sent data are encrypted with DRTS, they are semantically secure. The transmitted data cannot be accessed by $\mathcal{A}$ even if it is eavesdropped. Second, the ciphertext result (obtained by executing **DCP**, **DMP**, **DMAX**, **DMAX_n**) can be eavesdropped by $\mathcal{A}$ when transmitted between ESPs. These data are transmitted in ciphertext, which is not accessible to adversary $\mathcal{A}$ because of the semantic security of the encryption algorithm. Even if $\mathcal{A}$ corrupts multiple ESPs, the key *SK* cannot be recovered due to the properties of Shamir secret sharing. Even if the private key is recovered by corrupting more than $k$ ESPs (not both $\text{ESP}_\alpha$ and $\text{ESP}_\beta$ in the block), the original message is blinded due to the addition of random numbers to the plaintext, and $\mathcal{A}$ still cannot obtain useful information. The final result is sent to SR after re-encryption, and $\mathcal{A}$ is unable to decrypt the ciphertext of challenging SR in case $\mathcal{A}$ has the private keys of other users (i.e., not challenging SR's). The private keys of SR are chosen independently at random.

## 6. Efficiency Analyses

We analyze the computational complexity and communication overheads of the proposed five constructing blocks as well as the random forest evaluation scheme. In the scheme designed in this paper, the SR operation is encrypted before the data is sent and the RR performs only one decryption operation. Therefore, only the performance analysis of ESP is covered below.

*6.1. Analyses of Constructing Blocks*

Among these underlying protocols, ESPs can be divided into three categories: $\text{ESP}_\alpha$ that provides input in the protocol and processes the input; $\text{ESP}_\beta$ as a collaborator randomly selected by $\text{ESP}_\alpha$ each time the protocol is executed, and a residual server participant with a share of the master private key $\text{ESP}_i$.

6.1.1. Analysis of Computation Complexity

$\text{ESP}_\alpha$ adds a random value to the data to hide the original data to get a new ciphertext, and it needs to perform an exponential operation to process the new ciphertext obtained $(A,B)$, send $A$ and $B^{(\alpha)}$ obtained by executing **PDec** to a randomly selected $\text{ESP}_\beta$, and send $B$ to all ESPs.

The operations performed by $\text{ESP}_\beta$ in the first four protocols are receiving the data from each ESPs and execute **CDec**, processing the plaintext obtained from decryption according to the protocol, encrypting the processing result and sending it to $\text{ESP}_\alpha$. The

costs of plaintext processing is much less than the operations on ciphertext. Therefore, we can ignore the operation of plaintext in our analysis. In the re-encryption protocol, $\text{ESP}_\beta$ only performs **Recon** operation after accepting the data. $\text{ESP}_i$ accepts the data from $\text{ESP}_\alpha$ and executes **PDec**, and finally sends the result to $\text{ESP}_\beta$.

### 6.1.2. Communication Overheads

Ciphertext consists of two parts $[m_i] = (A, B)$. The bit length of the ciphertext is linearly related to the bit length of $N^2$. Thus, the length of each part of the ciphertext is $2\|N\|$ and the length of $[m_i]$ has $4\|N\|$.

We divide the operations of ESP into the most basic exponential and multiplicative operations, and summarize their computation complexity and communication overheads of each scheme in Table 3.

**Table 3.** Complexity analysis.

| Roles | Protocol | Computations | Communication Overhead |
|:---:|:---:|:---:|:---:|
| | **DMUP** | $2\textbf{Enc} + 6Exp + 10Mul$ | $t \cdot \|N^2\|$ |
| | **DCP** | $\textbf{Enc} + 6Exp + 4Mul$ | $t \cdot \|N^2\|$ |
| $\text{ESP}_\alpha$ | **DMAX** | $4\textbf{DMUP} + \textbf{DCP} + 2Exp + 2Mul$ | $t \cdot \|N^2\|$ |
| | **DRE** | $Mul + Exp$ | $t \cdot \|N^2\|$ |
| | **DMUP** | $2\textbf{CDec} + \textbf{Enc}$ | $\|N^2\|$ |
| | **DCP** | $\textbf{CDec} + \textbf{Enc}$ | $\|N^2\|$ |
| $\text{ESP}_\beta$ | **DMAX** | $9\textbf{CDec} + 5\textbf{Enc}$ | $\|N^2\|$ |
| | **DRE** | **Recon** | $\|N^2\|$ |
| | **DMUP** | $2\textbf{PDec}$ | $\|N^2\|$ |
| | **DCP** | **PDec** | $\|N^2\|$ |
| $\text{ESP}_i$ | **DMAX** | $9\textbf{PDec}$ | $\|N^2\|$ |
| | **DRE** | **PDec** | $\|N^2\|$ |

The computational complexity and communication overheads of **DMAX_n** relate to the amount of data $n$ is comparable as performing $n$ times **DMAX**.

### 6.2. Analysis of Proposed Privacy Preserving Random Forest

We have tested the performance of the scheme using a laptop with an Intel Core i5-7300HQ CPU 2.5 GHz and 24 GB RAM. Note that the encryption scheme used in this paper does not affect the results of any computation, i.e., it is consistent with the computation base on plaintext. Therefore, the classification accuracy is not considered in the performance analysis. Each server has a different number of models and structure, and all servers have only one decision tree that is a full binary tree. For $\text{ESP}_\alpha$, $\text{ESP}_\beta$ and $\text{ESP}_i$ are its roles, either sequentially or simultaneously, depending on the performance of the server. In the proposed scheme, $\text{ESP}_\gamma$ is the one selected from the servers responsible for aggregation, sorting and re-encryption of results. It waits for all ESPs to start running after performing the evaluation of the decision tree. We focus on the following three tests to demonstrate the consumption of a single ESP. Although only one server is tested, it is also assumed that the CP consists of ten ESPs with a threshold value of 5. These will be small impact on the **CDec** and aggregation.

Test 1: The depth $d$ of decision tree. Our experiments are complete trees, so there are $2^{(d-1)} - 1$ non-leaf nodes. We assume that module sizes $\|N\| = 1024$ and the resulting

categories are $|\mathcal{Z}| = (3, 6, 9, 12, 15)$. We perform the test by adjusting $d$. The results of the test are shown in Figure 3.

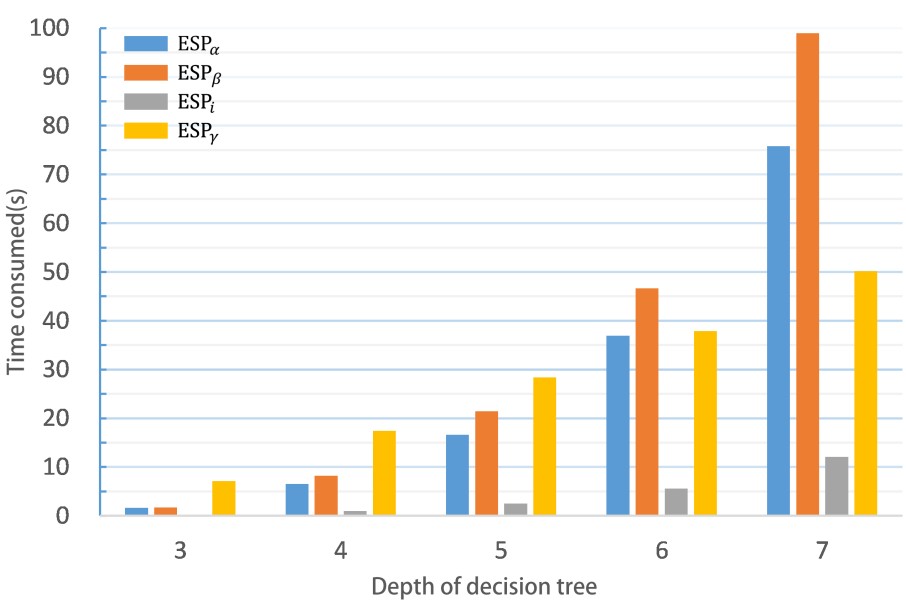

**Figure 3.** Effect of decision tree depth $d$ on the computation time of ESPs.

Test 2: The number of result categories. In this test, we set the depth of the tree is equal to $d = (5, 6)$ and the module sizes $\|N\| = 1024$. We perform the test by adjusting $|\mathcal{Z}| = (8, 12, 16)$, and the results of the test are shown in Figure 4.

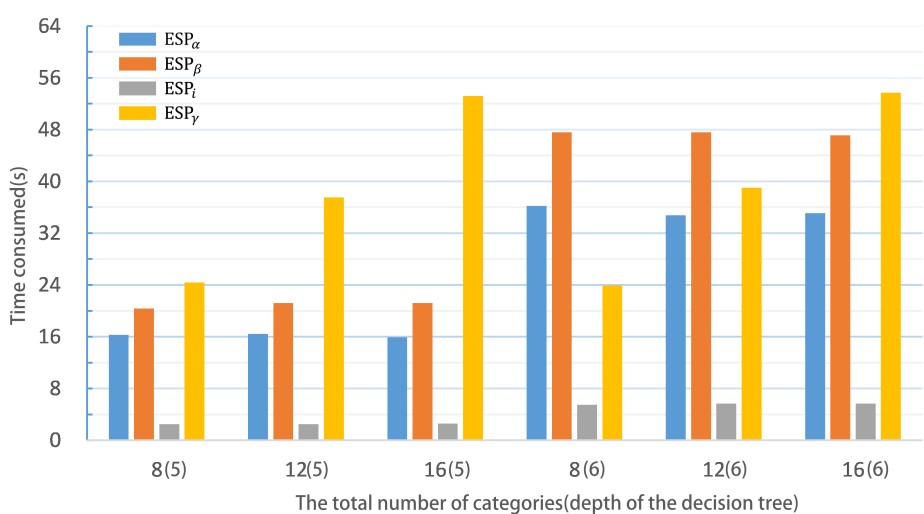

**Figure 4.** Effect of the total number of categories on the computation time of ESPs.

It can be observed from Figures 3 and 4 that the time consumption of $\text{ESP}_i$, $\text{ESP}_\alpha$ and $\text{ESP}_\beta$ is positively correlated with the number of nodes, while $\text{ESP}_\gamma$ is influenced by the number of types of evaluation results.

Test 3: The length of $N$. We set the tree depth in the scheme to $d = 6$ and the number of categories to $|\mathcal{Z}| = 15$. We choose different module sizes $\|N\|=(768,1024,1280,1536,1792,2048)$ for the test. Figure 5 shows the test results.

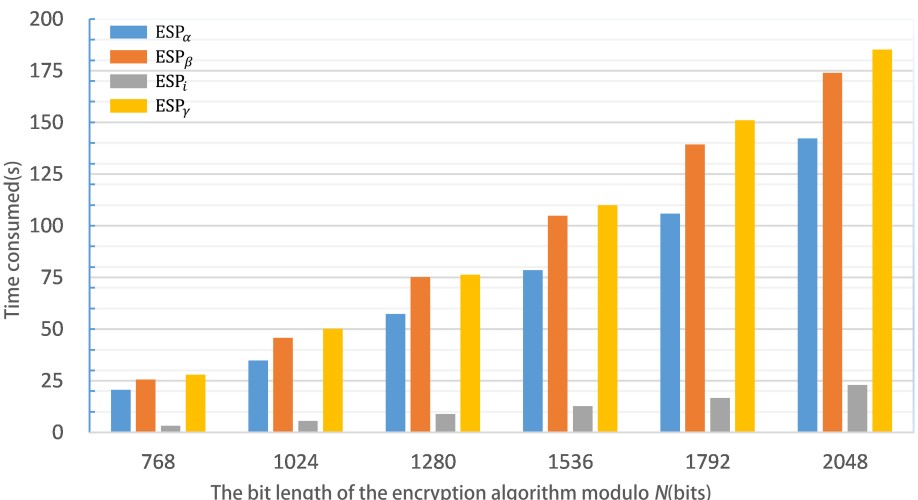

**Figure 5.** Effect of bit length of modulo *N* on the computation time of ESPs.

As described before, the individual roles are not executed sequentially, so in practice they can be executed synchronously or asynchronously depending on the situation. Moreover, as many random values are needed in the block, in practice the evaluation time of the random forest can be reduced by selecting random values in advance.

*6.3. Comparison with Existing Works*

To the best of our knowledge, most recent works focus on privacy-preserving decision trees [20,34,35]. However, none of these works can protect the training model and do not support users to be offline. The work in [36] is capable of solving the above problems by using a twin-cloud architecture. This scheme requires a KGC and requires both clouds to always stay online and these clouds are assumed not to collude. The work in [21] proposes a privacy preserving random forest evaluation based on FHE. Using the properties of MK-BGV, the model and data are encrypted and outsourced to the server for evaluation, which can resist collusion of cloud servers. However, it requires everyone to participate in the decryption process at the end (the server providing the model and the user providing the data). Compared with these solutions, our solution not only supports users to be offline and protects the training model, but also tolerates a small number of servers to be absent. Moreover, the evaluation results can be re-encrypted to the designated recipients. Because additive homomorphic encryption is used, it also has efficiency advantages over schemes based on FHE.

**7. Conclusions**

We propose a practical distributed and privacy preserving random forest evaluation scheme with fine grained access control. It not only protects the user's inputs and the server's model, but also realizes access control on the final evaluation results. Our scheme allows some users to be offline, and it can still be executed properly. Recently, many countries have issued laws and regulations to protect users' private information, such as GDPR in the EU and HIPPA in the US. Therefore, privacy preserving random forest evaluation can fulfil this requirement and it has more potential applications in real-world use, e.g., in healthcare and credit assessment.

In this paper, we have only considered the privacy-preserving evaluation using plurality voting, which may not suit for all random forest scenarios, and we hope to investigate some other methods in the future, such as majority voting, weighted voting and soft voting, so that it can be applied in a wider range of scenarios.

In addition, in our future work we plan to improve the secure comparison protocol to further reduce its computation overheads. In order to achieve high security level,

one needs to use a large modulo in the encryption algorithm, but this causes high computation overheads. In the future, we will explore homomorphic encryption schemes under other mathematical structures, and this could also contribute to a more efficient privacy-preserving random forest evaluation scheme.

**Author Contributions:** Conceptualization, Y.Z., H.S. and M.Z.; methodology, Y.Z.; software, Y.Z.; validation, H.S.; formal analysis, M.Z.; investigation, H.S.; resources, Y.Z.; data curation, Y.Z.; writing—original draft preparation, Y.Z.; writing—review and editing, H.S. and M.Z.; visualization, Y.Z.; supervision, M.Z.; project administration, M.Z.; funding acquisition, H.S. and M.Z. All authors have read and agreed to the published version of the manuscript.

**Funding:** This research was funded by National Natural Science Foundation of China under grants 61702168, 62072134, U2001205, and the Key Research and Development Program of Hubei Province under Grant 2021BEA163.

**Institutional Review Board Statement:** Not applicable.

**Informed Consent Statement:** Not applicable.

**Data Availability Statement:** Not applicable.

**Conflicts of Interest:** The authors declare no conflict of interest.

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
