# Peer review of "A Distributed and Privacy-Preserving Random Forest Evaluation Scheme with Fine Grained Access Control"

_symmetry, doi:10.3390/sym14020415_

Round 1
Reviewer 1 Report
This is a very interesting article written in a very clear presentation style. It seems your contribution mainly comes from ideas or extension of reference 10 & 12. However, there are few minor points and one major point to be noted.
1) You have used a very good analogy describing the flow of your model and assumed "Although a weak model may generate an unsatisfactory prediction, the other models can be used to balance the distortion" in line 28, I do not agree to this because "the other models" may have the same weakness that lead to group-think phenomenon, eventually providing a poorer result, i.e. biased on one side. In addition, in line 263, you have mentioned "If a minority of servers do not participate in the protocol due to network failure, the remaining servers can execute the protocol and output the evaluation results". How to define "minority" and is there any detection mechanism to below-minority level? There are some contradiction and flaws in your model and that affect the validity of your model. The choice and control of ESP is very important. This must be redesigned before publication acceptance process.
2) In line 241, you have assumed "honest-and-curious" but in line 367, you have also mentioned "semi-honest", are they the same?
3) In line 16, you should provide figures and reference supporting your saying of "increasing number of users".
4) Although this article was written in a clear and systematic way, there are few grammatical mistakes, e.g. "corporate" should be changed to "incorporate" in line 58 & 247.
Reviewer 2 Report
The paper is devoted to the application of a random forest evaluation scheme based on asymmetric encryption which can be used in a variety of applications.
The paper is written in good English, thus no extensive language corrections are foreseen.
The paper is well-referenced with mentioning the state-of-the field-related sources.
The illustrated results are good and the overall soundness of the paper is enough to be published in Symmetry Journal.
Together with that, I'd like to propose several improvements which will increase the overall paper quality from my point of view:
- The conclusions section can be enhanced (or a separate section "Discussion" can be inserted) thus allowing the reader to be acquainted not only with a short summary of the paper but with the future research and development directions and improvements what is more important as well;
- What are the measures in Figures 3-5? X and Y axis?
- The application scenarios and possible use cases of the proposed scheme should be addressed
Round 2
Reviewer 1 Report
I have just reviewed the second version and still found the problem in the group-think possibility, i.e. no value in this research which aims to have objective contribution. There are two methods that the authors can address this important fundamental research problem. The first one is to design a separate method (e.g. assessment scheme for the ESP with weights assigned showing their trustworthiness) to get rid of this possibility. The second method is much simpler by stating this is the weakness of this research in the conclusion section and hope this assessment or similar countermeasure can be done in future.
Round 3
Reviewer 1 Report
I am now satisfied with the modified conclusion.